# Views of the Future of Partners of People with Multiple Sclerosis Who Attended a Lifestyle Modification Workshop: A Qualitative Analysis of Perspectives and Experiences

**DOI:** 10.3390/ijerph18010085

**Published:** 2020-12-24

**Authors:** Sandra L. Neate, Keryn L. Taylor, Nupur Nag, George A. Jelinek, Steve Simpson-Yap, William Bevens, Tracey J. Weiland

**Affiliations:** 1Neuroepidemiology Unit, Centre for Epidemiology and Biostatistics, Melbourne School of Population and Global Health, The University of Melbourne, Parkville 3010, Australia; keryn.taylor@unimelb.edu.au (K.L.T.); nnag@unimelb.edu.au (N.N.); g.jelinek@unimelb.edu.au (G.A.J.); steve.simpsonyap@unimelb.edu.au (S.S.-Y.); william.bevens@unimelb.edu.au (W.B.); tracey.weiland@unimelb.edu.au (T.J.W.); 2Department of Psychiatry and Psychosocial Cancer Care, St Vincent’s Hospital Melbourne, Fitzroy 3065, Australia

**Keywords:** multiple sclerosis, partners, lifestyle modification, future, confidence

## Abstract

People with multiple sclerosis (PwMS) often experience uncertainty and fear about their futures. Partners of PwMS may share their concerns and experience fears about their own futures, limitations on their lives, ability to work, and becoming a carer. For PwMS, modification of lifestyle-related risk factors has been associated with improved health outcomes. For PwMS who attended residential lifestyle modification workshops (RLMW), sustained improved health outcomes have been demonstrated. Whether improved outcomes for PwMS who engage with lifestyle modification translate to improved partner perceptions of the future, is yet to be explored. We explored the perspectives of partners of PwMS who had attended a RLMW and the impact that the person with MS’s illness and their engagement with lifestyle modification had on their partners’ views of the future. Analysis of 21 semi-structured interviews used a methodology informed by Heidegger’s Interpretive Phenomenology. Three themes emerged: ‘uncertainty’, ‘planning for the future’ and ‘control, empowerment and confidence’. Subthemes included MS and lifestyle modification being a catalyst for positive change; developing a sense of control and empowerment; and hope, optimism and positivity. Lifestyle modification may provide benefits, not only to PwMS, but also to their partners, and should be considered part of mainstream management of MS.

## 1. Introduction

Multiple sclerosis (MS) is an autoimmune, degenerative, demyelinating condition of the central nervous system with a highly variable disease course. Because of the unpredictability of the illness, people with MS (PwMS) often experience concerns regarding whether, when, and how symptoms will develop and progress [1]. Uncertainty about the future is extremely common for PwMS and may cause fear, anxiety, and pose a threat to the person’s identity [2].

MS also has a significant impact on wellbeing and quality of life for family members, often creating psychological stress [3]. Understandably, intimate partners of PwMS may experience uncertainty and worry about the PwMS and also about potential limitations to their own lives, including their careers and social lives [4,5] and, ultimately, the prospect of a becoming a carer [5,6] and the associated changes in their identity [4]. This unpredictability may cause feelings of powerlessness for the partner and a sense of being unable to control their future [7]. For partners in caring roles the diverse range of manifestations and unpredictable progression of MS cause further practical and psychological challenges [8]. 

Associations between health outcomes and lifestyle-related risk factors in MS, such as diet, smoking, lack of exercise, inadequate vitamin D and sunlight, and stress, have been identified [9,10,11]. Healthy lifestyle behaviours that reduce risk factors have been associated with reduced fatigue [12], decreased depression risk [13], relapse rate reduction [14], and decreased disability [9,10,11,15]. Longitudinal evaluations of intensive residential workshops promoting lifestyle-related risk factor modification for PwMS have demonstrated sustained improved health outcomes three and five years following the workshop, and an ability to maintain the modifications over the short to medium term [16,17,18]. 

For PwMS who adopt intensive lifestyle modification, the changes undertaken and the potential improved outcomes may understandably affect their partners. If partners choose to adopt similar modifications themselves, to support the person with MS and potentially improve their own health, the impact on the partners’ lives may be considerable. For those partners of PwMS who have engaged with lifestyle modification, the ways in which partners perceive their futures, their aspirations, goals and confidence in the future, has not been explored. 

We aimed to better understand the views and experiences of partners of a sample of PwMS who had engaged with lifestyle modification through attendance at a residential lifestyle modification workshop (RLMW). This paper reports the fourth over-arching theme developed from this qualitative analysis, three of which have previously been reported [19,20,21], and explores partners’ views of the future and the impact that both MS and engagement with lifestyle modification may have had on these views. 

## 2. Materials and Methods 

The methodology of this qualitative interview study has been previously described in detail [19], and is described in an abbreviated form below.

### 2.1. Study Design

The philosophy of Heidegger’s interpretive phenomenology [22] guided this study. This design allowed researchers to interpret participants’ experiences within their own contexts while bringing researchers’ knowledge and experience to the interpretation.

### 2.2. The Residential Lifestyle Modification Workshop (RLMW) 

The three-to-five-day RLMWs (the workshops) relevant to this study, which PwMS and sometimes partners attended, were held in Australia, New Zealand, United Kingdom and Europe between 2012 and 2016. The workshops, previously described in detail [16], consisted of interactive and experiential sessions delivered by healthcare professionals (mostly doctors with MS). They provided best available evidence on modification of lifestyle-related risk factors including smoking cessation, diet, exercise, sunlight and vitamin D, and stress reduction techniques. Attendees were able to discuss, reflect, and share experiences and develop close connections with others. 

### 2.3. Participant Recruitment

Participants were partners of PwMS. The PwMS were from an international online cohort of 2466 participants in the Health Outcomes and Lifestyle In a Sample of people with Multiple sclerosis (HOLISM) study [23]. Those in the HOLISM cohort who indicated in their surveys that they had attended a RLMW (*n* = 345) and that they were partnered (*n* = 280) were selected from the dataset. These PwMS were then randomised and emailed in groups of 10 and asked to forward the email to their partner if agreeable. The email invited the partner to provide consent to participate and complete a short survey of demographic data, details of the workshop the person with MS (and potentially they) attended, and names of the facilitators if known. A ‘No’ response from the partner concluded the survey. Reasons for declining participation were not explored. A ‘Yes’ response led to contact by researchers to organise an interview with the partner. Interviews and analysis occurred concurrently, enabling researchers to identify when apparent data saturation occurred [24] and, at that point, further recruitment ceased. 

Interviewed partners were from Australia (eight), New Zealand (six), the United Kingdom (six) and Switzerland (one). The researchers considered that the somewhat heterogeneous cultural backgrounds of participants enriched the data. Participant recruitment is described in Figure 1. 

### 2.4. The Interviewers

The two female specialist medical practitioner interviewers (SN and KT) had independently facilitated the workshops. They believed their extensive experience in clinical and research interviews and in workshop facilitation would benefit communication with participants and was consistent with Heidegger’s philosophy of the validity of interviewers’ experiences being brought to data collection and interpretation [25]. Interviewers introduced themselves and their role in the research during the interview. Interviewers ensured they had not facilitated the workshops that the person with MS (and potentially the participant) had attended to avoid a prior relationship.

### 2.5. The Interview

The semi-structured interview (Appendix A) was developed following an examination of existing literature of partners’ experiences of MS and from prior researcher knowledge of workshops. Partners were invited to explore the impact of both MS and lifestyle modification.

Partners were interviewed in their homes by telephone or Skype between July and October 2016. Interviews were recorded, stored in password-protected files, de-identified, and transcribed by an independent company. 

### 2.6. Analysis

The researchers used a hermeneutic interpretive process to analyse and interpret the data [22]. Researchers believed their knowledge and experience was an essential component of the research, but acknowledged the potential impact of their preconceptions on the conduct of interviews and data interpretation. Analysis was conducted without a predetermined coding structure. Interviews, transcription, and coding were conducted simultaneously. Each interview was repeatedly read, codes were identified (SN), aggregated into categories, then broader themes were developed that explored relationships between the categories and reflected researchers’ understanding of the meaning of participants’ reflections (SN, KT, and TW). 

Following the analysis of the complete dataset, four over-arching themes were identified. Due to the richness of the data, each over-arching theme was then analysed separately. This paper reports partners’ views of the future and the impact that both MS and engagement with lifestyle modification following attendance at a workshop may have had on these views. 

The research aimed to explore partners’ experiences following the single intervention of the attendance at a workshop by the person with MS. The researchers did not evaluate the degree of lifestyle modification adopted following the workshop by either the person with MS or the partner, however, interviews revealed that all PwMS had made significant lifestyle changes. There was no requirement that partners had attended the RLMW. Some interviewed partners had attended the workshop with the PwMS. Researchers did not aim to differentiate between those partners who had (10/21 participants) and had not (11/21 participants) attended the workshop and considered that the different experiences of participants added to the breadth of their perspectives. 

### 2.7. Rigour

Independent transcription and checking of transcriptions with recordings (SN) ensured accuracy of the data. Verbatim quotations assisted transparency [26]. Nvivo software outputs demonstrated coding and theme evolution [27]. Frequent rereading of manuscripts, thematic development by multiple researchers and researcher reflective discussions enhanced credibility [28]. Memos of researcher meetings recorded thematic development and researcher self-reflections to assist reflexivity [29]. The COnsolidated criteria for REporting Qualitative research (COREQ) checklist [30] was used to ensure rigour. 

### 2.8. Ethics Approval

All subjects gave their informed written and verbal consent for inclusion before they participated in the study. The study was approved by the University of Melbourne Human Research Ethics Committee (ID number 1545280.1).

## 3. Results

Twenty-one interviews were conducted. Interview duration ranged from 20 to 62 min (average 36 min). All participants indicated during the interviews that the PwMS had adopted the lifestyle modifications recommended during the workshop, although the degree of adoption was not specifically queried. Many partners indicated they had also adopted changes. Lifestyle modification, and the observed changes in the person with MS, was implicit in many participants’ comments. Others explicitly spoke of the impact. 

### 3.1. Characteristics of the Person with MS and Their Partners (Participants) 

Most participants were male (71%). More than half were aged over 50 years (57%), had a relationship of more than 20 years’ duration (52%), and had attended the workshop (52%). Most (62%) were employed. Regarding the people with MS, one third (33%) had a partner reported diagnosis of relapsing remitting MS and 28% had progressive MS; 57% had attended the workshop in the preceding five years; 66% had a diagnosis duration of 10 years or less and 76% had no disability. All relationships in the interviewed partners were heterosexual. Characteristics of the partners and the PwMS are reported in Table 1.

### 3.2. Themes

Participants reflected on both the impact of MS and the ways in which engagement with lifestyle modification had affected their views of the future. Three major themes were identified, ‘uncertainty’, ‘planning for the future’ and ‘control, empowerment and confidence’. Unnecessary words were deleted from quotations. Participants refer to ‘OMS’, the Overcoming Multiple Sclerosis program, the evidence-based synthesis of the literature [31] delivered during the workshops described. Anonymised participant numbers are reported as P-number. F = female, M = male.

#### 3.2.1. Uncertainty

The theme of uncertainty explored the expressions of doubts and fears for the future that MS had caused and how partners experienced and addressed those fears and uncertainties. Partners recognised the health of the PwMS was unpredictable, despite working hard on lifestyle modification, and expressed how they were worried by the unknown.


*It’s about the unknown…how we might manage if he does reach a point where he is not able to walk. I’m hoping I just don’t die. (P3, F)*


Some expressed they could never be free of the anxiety that uncertainty caused them. MS was always at the back of their minds as they faced the future.


*It’s still a demon that’s in your life. (P21, M)*


Some partners imagined future outcomes and then realised that, as MS was unpredictable, these predictions were futile.


*It’s so unpredictable. I need to stop trying to predict what’s going to happen (P7, F)*


When the PwMS had remained well and some degree of confidence in the future had begun to develop, any small event or change in the person’s health could cause uncertainty to return without warning and undermine confidence.


*But then every now and again you’ll hit a small bump that brings you back to the reality of…living with MS every day. (P21, M)*


Some described strategies, such as acceptance and dealing with change only when it occurred. Some found taking ‘one day at a time’ and not thinking beyond the short-term effective.


*So we are more concerned about just (adopting) a day by day, month by month kind of approach to life. (P9, M)*


Other partners seemed less concerned regarding potential uncertainty. They recognised many aspects of life were uncertain; that MS was no different and they just had to deal with challenges when they arose.


*To be honest, I have not thought about [partner] being incapacitated in any way. I don’t know if that, or when that, might happen, but it’s something that we would handle when it does happen. (P11, M)*


#### 3.2.2. Planning for the Future

This theme explored how MS and lifestyle modification influenced partners’ plans. At times, anxiety-driven uncertainty regarding the future significantly influenced decision-making, leading to sense of urgency in making major life decisions. Partners described MS as a ‘member of the family’ that had to be considered when making decisions.


*We sort of have the MS thing. It’s part of the decision…Yeah, the fourth person in the family. (P5, M)*


For some, even successful self-management and continuing good health was not sufficient to deliver confidence in the future. Some partners changed their lives despite the observed good health of the person, and described ‘taking out insurance for the future’.


*We’ve made changes in our lives...Even in terms of how we spend money, we’re more frugal I guess because there is a possibility that in 15 years time…(P5, M)*


For some, the perceived unpredictability of the future added urgency to making major life decisions, such as having children.


*It’s definitely changed our plans…One of the things is that we planned on having children much later on life…we ended up having [child] much earlier. (P7, F)*


Others felt a similar need to change plans but, rather than being anxiety-driven, saw the uncertainty as an opportunity to make proactive decisions that allowed freedom to undertake the things they enjoyed.


*In fact, I used it as a reason—I retired about a year (earlier)...and we took quite a few overseas trips (P12, M)*


Some described moving further away from anxiety-driven urgency and described a new perspective, where uncertainty became a catalyst for making plans that may, in other circumstances, have been delayed. They experienced a sense of wanting to do things immediately and to not postpone plans.


*We’re not going to live life waiting. If we really want to do it we’re not going to wait until we’re retired to do it. We’ll do it now...so there’s a bit more urgency I guess, which isn’t a bad thing. (P5, M)*


For some MS became an incentive to plan for a different future that included new and exciting challenges to extend themselves beyond what they previously would have undertaken.


*She would never have travelled to Central America and backpacked pre-MS. It’s almost like we’re saying MS isn’t holding us back in any way. (P5, M)*


#### 3.2.3. Control, Empowerment and Confidence

This theme explored a shift from worry about an uncertain future towards a sense of confidence in and control of the future. Some partners were able to let go of previous doubts and fears due to the continued health of the PwMS and described a shifting perspective.


*I think the future is looking very good. Interviewer: So those initial fears and uncertainties are they still lurking there somewhere? P9: I think they’ve gone. (P9, M)*


MS no longer influenced every decision, and freedom to make decisions without undue consideration of MS developed. This shift developed alongside engagement with lifestyle modification and with it came a sense of empowerment and that management of MS and the future was more within their control.


*That’s what I love the most about OMS and that’s what really appeals to me more than anything else, was the fact that you were in control. (P7, F.)*


The new confidence allowed them to plan positively for the future, rather than feeling unable to plan or feeling a sense of urgency because of unpredictability.


*MS certainly isn’t stopping us from planning for the future...We’re so far at the opposite end of that spectrum. (P21, M)*


As well as having feelings of control, some partners felt enough confidence to make major financial commitments that they would previously not have undertaken.


*I remember saying when we bought this house. We would never have gone and done all of that (taken a mortgage) if we weren’t confident in paying it off. (P1, F)*


From the sense of control, empowerment and confidence came positivity. Some partners described feeling back to where they had been before MS and, in some cases, even better than before MS.


*Really completely positive. I guess I feel how I felt when we got married before we found out about MS, what I thought our lives were going to be like, like that is just how I see our lives now and better in a way than how I would have felt then. (P20, F)*


## 4. Discussion

Most explorations of the impact of MS on partners’ perceptions of the future, have been limited to partners of PwMS who adopted conventional medical management, particularly those anticipating or undertaking care-giving roles [8,32,33,34]. Participants in our study were unique in being partners of PwMS who had engaged with lifestyle modification through attendance at a workshop promoting and supporting evidence-based healthy lifestyle behaviours for PwMS. The themes described in our study progressed from the sense of uncertainty and even fear that MS engendered, towards confidence in the future, through accepting and at times embracing changes to their lifestyles.

Some of the themes reported by our partners were similar to themes described in the literature by partners of PwMS who adopted conventional medical management. This was particularly true of the uncertainty that the unpredictable course of MS caused. Despite the PwMS having engaged with lifestyle modification, some partners still experienced uncertainty about what their futures held. In previous studies, partners described MS as ‘like living your life with a weight on your back all the time’ [4]. In our study, MS was at times described as a member of the family who had to be considered, something that was always at the back of the mind, and as a demon. Trying to live with unpredictability may be burdensome [4] and partners in our study also expressed these feelings.

However, in our study, some partners expressed that uncertainty represented an opportunity rather than being an impediment. Prior research has found that, for partners who were carers, opportunities arose for personal growth, to develop greater appreciation of life and to reprioritise what was important in life [34]. The ability to see benefits improved life satisfaction and other outcomes for partners [35]. Partners in these studies saw opportunities for personal development from facing adversities. Partners in our study differed in that they decided to undertake challenges they would not previously have considered possible, such as adventure travel, changing careers or retiring earlier and acted immediately rather than putting things off. These actions were driven by a newfound sense of hope and a wish to make the most of life and good health, rather than by fear or uncertainty.

Participants in our study described developing confidence in their futures, a concept rarely described in the MS literature. This confidence contrasted with the more commonly expressed worry and uncertainty and feelings of lack of control. Other studies reported partners developed confidence in the future by reflecting on the ways in which they had successfully overcome challenges in the past, prior to MS, to gain confidence in their ability to do the same in the context of MS [4]. However, the confidence described in other studies was limited to the ability to manage obstacles and challenges. In contrast, our study’s partners appeared to have genuine confidence about positive futures, based on observed positive health outcomes of the PwMS, rather than just an ability to manage challenges.

The partners in our study developed more than just feelings of hope and optimism. Some partners explicitly spoke of the impact of lifestyle modification and described a genuine sense of having regained control and feelings of empowerment to influence their futures that they attributed to lifestyle modification. Possessing a sense of agency or control of one’s life, also described as having a sense of mastery, is defined as ‘the extent to which one regards one’s life chances as being under one’s control rather than fatalistically ruled’ [36]. People’s belief in their own effectiveness is known to positively affect the ability to maintain behaviour change [37]. Therefore the partner’s greater sense of control and empowerment may ultimately help both people in the relationship to make and maintain lifestyle behaviours associated with improved quality of life. Therefore, in a parallel process, the sense of control that is achieved from improved outcomes may lead to a greater chance of maintaining healthy behaviour.

### Limitations

The views expressed by participants and their analysis by researchers, are unique to this exploration. Extrapolation to the wider MS community, those in countries other than Australia, NZ, UK and Switzerland, those in non-heterosexual relationships and those no longer in relationships, may differ from the interpretations in this study.

Some themes in this analysis shared similarities with themes identified in our previous studies. The researchers considered that ‘hope and optimism’ [19] and ‘control, empowerment and confidence’ were similar but sufficiently different concepts that warranted separate exploration. The theme of control and empowerment describes concepts related to the ability to self-manage and control health and life, important factors influencing the future of those living with MS.

Recruitment was randomised and the resultant numbers of participants who had and had not attended the RLMW were not predictable. Approximately half of our participants had attended the RLMW with the person with MS and half had not. Researchers did not analyse the difference between responses from attenders and non-attenders. Researchers considered the potential variations in perspectives would add depth and breadth to the analysis.

The unique group of participants, partners of PwMS who have attended a residential workshop and adopted lifestyle modification, were intentionally selected to explore how their experiences compared with or differed from those in the existing literature.

## 5. Conclusions

Partners in our study experienced some doubts and uncertainties common to those of other partners dealing with MS. However, lifestyle modification and its observable outcomes for the PwMS enabled some partners to develop a sense of empowerment and control, and a subsequent confidence and positivity about their futures. Such perspectives are novel and add support for the recommendation of modification of lifestyle-related risk factors in MS. Clinicians should consider lifestyle modification in the management of PwMS and their partners.

## Figures and Tables

**Figure 1 ijerph-18-00085-f001:**
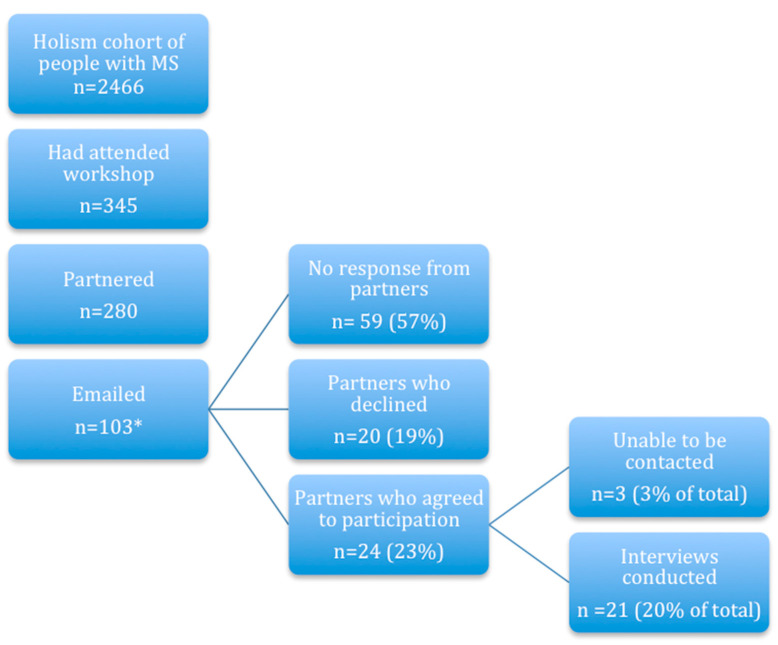
Participant recruitment. Total of 280 potential participants were randomised. * 103 were invited to participate in groups of 10; after 21 interviews had been conducted from 103 invitees and apparent data saturation, no further invitations were sent.

**Table 1 ijerph-18-00085-t001:** Demographic data of people with MS and partners (participants) as reported by participants.

People with MS				Partners				
* Type of MS	Years since Diagnosis	Years since Workshop	^ Disability	Sex	Age	Workshop Attendance with Person with MS	Employment	Years of Relationship with Person with MS
**RRMS**	5–10	>5	No	Female	30–39	Yes	Full time	11–20
**PPMS**	5–10	2–5	No	Male	20–29	Yes	Full time	1–10
**Unsure**	0–5	1–2	No	Female	40–49	No	Full time	21–30
**RRMS**	0–5	1–2	No	Female	50–59	Yes	Part time	21–30
**RRMS**	11–20	>5	No	Male	40–49	Yes	Part time	1–10
**SPMS**	11–20	>5	Yes	Female	70–79	Yes	Retired	>50
**SPMS**	11–20	>5	Yes	Female	60–69	No	Retired	41–50
**Unsure**	5–10	2–5	No	Female	40–49	Yes	Full time	1–10
**SPMS**	0–5	2–5	No	Female	20–29	Yes	Part time	1–10
**Unsure**	5–10	>5	No	Female	60–69	No	Full time	21–30
**PPMS**	11–20	>5	Yes	Male	60–69	Yes	Full time	41–50
**CIS**	0–5	2–5	No	Female	60–69	Yes	Retired	31–40
**RRMS**	5–10	>5	No	Male	40–49	No	Full time	1–10
**Unsure**	5–10	1–2	No	Female	30–39	No	Full time	1–10
**Unsure**	11–20	>5	No	Female	60–69	No	Retired	41–50
**Unsure**	0–5	1–2	No	Female	50–59	No	Unable	31–40
**RRMS**	0–5	1–2	No	Male	20–29	No	On leave	1–10
**PPMS**	5–10	2–5	Yes	Male	50–59	No	Full time	11–20
**RRMS**	>40	2–5	No	Male	60–69	Yes	Retired	41–50
**RRMS**	5–10	2–5	Yes	Male	60–69	No	Full time	11–20
**Unsure**	11–20	>5	Yes	Male	70–79	Yes	Retired	21–30

* Type of MS (partner reported); ^ Disability as defined by question: ‘Has the person with MS used a walking aid in the last 1 week?’; CIS = clinically isolated syndrome; leave = maternity leave; PwMS = people with multiple sclerosis; RRMS = relapsing remitting MS; PPMS = primary progressive MS; SPMS = secondary progressive MS; unable = had to stop work to care for person with MS.

## Data Availability

Restrictions apply to the availability of these data. Data was obtained from participants who were assured of confidentiality and are available from the authors with permission from participants.

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
