# Peer review of "Views of the Future of Partners of People with Multiple Sclerosis Who Attended a Lifestyle Modification Workshop: A Qualitative Analysis of Perspectives and Experiences"

_ijerph, 2020, doi:10.3390/ijerph18010085_

Round 1

Reviewer 1 Report

A clean and straightforward piece of research

Author Response

Thank you for your very positive review.

Reviewer 2 Report

This qualitative study addresses perceptions of the future among partners of people with multiple sclerosis (PwMS) who attended lifestyle modification workshops (RLMW). A sample of 21 partners of PwMS was involved, using semi-structured interviews. The themes “uncertainty”, “planning for the future”, and “control, empowerment and confidence” emerged from qualitative analyses following Heidegger’s Interpretive Phenomenology methodology.

A major concern relates to heterogeneity in this sample. Participants were from Australia, NZ, UK and “Europe”. However, the authors do not specify how many participants were from each area. Which European countries participated in this project should also be clarified. Cultural issues as well as cross-national differences in policies pertinent to health and formal and informal caregiving might affect perceptions of the future in partners of PwMS.

Furthermore, some participants attended the RLMW themselves, while others did not. The authors state that how partners’ attendance to the RLMWs affected partners’ views was outside the scope of this study. However, partners who participated in the RLMWs and partners who did not cannot be considered comparable, without first proving that participation had no effects on views of the future. A specific question on this should have been included in the interview, as to exclude any confounding role of participation in the RLMW.

Finally, this is potentially the fourth publication attached to the same data. The authors clearly justify this by stating that four overarching themes emerged from interviews, and each was then considered separately. Yet, I wonder whether this theme could have been presented along with a similar one, i.e., psychological shift experienced by partners of PwMS (which also included hope and optimism).  

Author Response

Thank you very much for your comments which I have addressed below:

1) A major concern relates to heterogeneity in this sample. Participants were from Australia, NZ, UK and “Europe”. However, the authors do not specify how many participants were from each area. Which European countries participated in this project should also be clarified. Cultural issues as well as cross-national differences in policies pertinent to health and formal and informal caregiving might affect perceptions of the future in partners of PwMS.

The workshops were held in Australia, New Zealand, the UK and Europe between 2012 and 2016. Most were held in Australia and New Zealand during that time. One was held in Austria in 2015 and approximately six were held in the UK. Of the partners to whom we spoke, 8 were from Australia, 6 were from the UK, 6 were from NZ and one was from Switzerland.

The researchers considered that cultural differences were not significant considering the cohort were mostly from Australia, NZ and the UK and that health policies were relatively similar in these areas also. However, the researchers also considered that some heterogeneity would add interest to perceptions, and generally this is encouraged in qualitative research.

I have specified the nationalities of the participants and addressed your concerns as follows"

" Interviewed partners were from Australia (eight), New Zealand (six), the United Kingdom (six) and Switzerland (one). The researchers considered that the somewhat heterogenous cultural backgrounds of participants enriched the data."

2) Furthermore, some participants attended the RLMW themselves, while others did not. The authors state that how partners’ attendance to the RLMWs affected partners’ views was outside the scope of this study. However, partners who participated in the RLMWs and partners who did not cannot be considered comparable, without first proving that participation had no effects on views of the future. A specific question on this should have been included in the interview, as to exclude any confounding role of participation in the RLMW.

Thank you. This is an interesting point. When we recruited we did not know whether participants who consented to participation had attended the RLMW themselves. We were interested to realise that half of them had attended. And of course, this did not become apparent until recruitment was complete. The researchers, however did not consider that it would be a confounding issue if some participants had attended and some hadn't. We had not intended their own attendance to be a focus of the interview, and it was not. It was the person with MS' attendance we were focused on and how this attendance affected the partners' views. The only direct discussion of the RLMW was the question regarding whether they had attended or not in the original email. Therefore we did not include a question about this in our survey. Our focus was very much on "how have the lifestyle modifications that the person with MS has adopted affected your life and your views of the future?"

We considered following recruitment that, as approximately half had attended the workshop, it may be interesting to examine the two groups of partners separately, but thought that comparing and contrasting participant responses may detract from the qualitative exploration, where we aimed to understand how people felt about lifestyle modification, whether or not they had been to the workshop, and that the different perspectives would add breadth to the research. 

I have changed the paragraph to attempt to further address your concern and modified the explanation in the methods:

The research aimed to explore partners’ experiences following the single intervention of the attendance at a workshop by the person with MS. The researchers did not evaluate the degree of lifestyle modification adopted following the workshop by either the person with MS or the partner, however, interviews revealed that all PwMS had made significant lifestyle changes. There was no requirement that partners had attended the workshop. Some interviewed partners had attended the workshop with the PwMS. Researchers did not aim to differentiate between those partners who had (10/21 participants) and had not (11/21 participants) attended the workshop and considered that the different experiences of participants added to the breadth of their perspectives. Interviewed partners were from Australia (eight), New Zealand (six), the United Kingdom (six) and Switzerland.

3) Finally, this is potentially the fourth publication attached to the same data. The authors clearly justify this by stating that four overarching themes emerged from interviews, and each was then considered separately. Yet, I wonder whether this theme could have been presented along with a similar one, i.e., psychological shift experienced by partners of PwMS (which also included hope and optimism).  

Thank you. The authors were keen for the reviewers and readers to understand that other publications from this study existed. When the data were analysed the researchers concluded there were four distinct overarching themes each of which provided a substantially different window into participants' experiences: "Psychological shift", that described the psychological and emotional changes experienced; "Taking active steps", that explored the concrete changes that people had incorporated in to their lives; "On the path together" that explored the impact that MS and lifestyle modification had had on the relationship between the PwMS and their partners; and "Views of the future" where participants expressed their attitudes and feelings towards their futures. These overarching themes were then analysed sequentially, so it is true that some of the sub-themes that emerged were related between the overarching themes. Once again, this only became apparent after analysis of sequential overarching themes. However, the researchers considered that sub-themes remained sufficiently different as to warrant separate reporting.

While hope and optimism, expressed in the psychological shift paper, and control and empowerment in this paper, are related concepts, the researchers considered that they are sufficiently different concepts. Hope and optimism considered the emotional shifts and experiences that had been incorporated within the partners. The theme of control and empowerment goes more to the concepts of self-efficacy and self-management, the sense of having control over one's life, important concepts in the management of MS. 

I have modified the last paragraph of the discussion to highlight this difference:

The partners in our study developed more than just feelings of hope and optimism. Some partners explicitly spoke of the impact of lifestyle modification and described a genuine sense of having regained control and feelings of empowerment to influence their futures that they attributed to lifestyle modification. Possessing a sense of agency or control of one’s life, also described as having a sense of mastery, is defined as “the extent to which one regards one’s life chances as being under one’s control rather than fatalistically ruled”[37]. People’s belief in their own effectiveness is known to positively affect the ability to maintain behaviour change [38]. Therefore the partner’s greater sense of control and empowerment may ultimately help both people in the relationship to make and maintain lifestyle behaviours associated with improved quality of life. Therefore, in a parallel process, the sense of control that is achieved from improved outcomes may lead to a greater chance of maintaining healthy behaviour.

I have added the following into the limitations to address several of the issues raised:

The views expressed by participants and their analysis by researchers, are unique to this exploration. Extrapolation to the wider MS community, those in countries other than Australia, NZ, UK and Switzerland, those in non-heterosexual relationships and those no longer in relationships, may differ from the interpretations in this study.

Some themes in this analysis shared similarities with themes identified in our previous studies. The researchers considered that “hope and optimism” [19] and “control, empowerment and confidence” were similar but sufficiently different concepts that warranted separate exploration. The theme of control and empowerment describes concepts related to the ability to self manage and control health and life, important factors influencing the future of those living with MS.

Recruitment was randomised and the resultant numbers of participants who had and had not attended the RLMW were not predictable. Approximately half of our participants had attended the RLMW with the person with MS and half had not. Researchers did not analyse the difference between responses from attenders and non-attenders. Researchers considered the potential variations in perspectives would add depth and breadth to the analysis.

Reviewer 3 Report

The paper is on a very interesting paper, but there are some aspects that need more detailed description. From my opinion, it is not clear how to evaluate the effect of the workshop on the person with sclerosis Multiple without a control group and/or with an assessment of some kind of social desiderability of the replies. 

It is not clear what does it means the follow: "Participants were partners of PwMS from an international online cohort of 2466 participants in the Health Outcomes and Lifestyle In a Sample of people with Multiple sclerosis (HOLISM) study [23]. Those in this cohort who had attended a RLMW (n=345) and were partnered (n=280) were randomised and emailed in groups of 10"

in conclusion, It is not clear what does it means the follow.Our study found that partners of PwMS who undertake lifestyle modification as part of MS management, experience some doubts and uncertainties common to those of other partners dealing with MS. However, lifestyle modification and its observable outcomes enabled some partners to develop a sense of empowerment and control, and a subsequent confidence and positivity about the future. Such perspectives are novel and add support for the recommendation of modification of lifestyle-related risk factors in MS. Clinicians should consider lifestyle modification in the management of PwMS and their partners."

I suggest to better describe aims, results and discussion of results. 

.

Author Response

Thank you very much for your valuable comments. The researchers acknowledge that it has been an on-going challenge of reporting results from this study to adequately and clearly describe the participants, the intervention and the outcomes. I will try to clarify these points.

1) The paper is on a very interesting paper, but there are some aspects that need more detailed description. From my opinion, it is not clear how to evaluate the effect of the workshop on the person with sclerosis Multiple without a control group and/or with an assessment of some kind of social desiderability of the replies. 

Our aim was not to evaluate the effect of the workshop on the people with MS. Our research group has undertaken quantitative studies to evaluate this. Our experiences of the workshop were, however, that not only the people with MS, but their partners, experienced significant changes to their lives subsequent to the workshop. As most of the experiences of partners, that have been previously described in the literature, have been negative, we aimed to explore this particular group of partners to see if their experiences differed. But there was no intention to evaluate the effect of the workshop on the person with MS. As this research is qualitative as opposed to quantitative, there is no requirement for a control group. Given the qualitative nature of the research, we were interested in examining the themes present in the full range of replies of the participants, whatever their nature or viewpoint. 

2) It is not clear what does it means the follow: "Participants were partners of PwMS from an international online cohort of 2466 participants in the Health Outcomes and Lifestyle In a Sample of people with Multiple sclerosis (HOLISM) study [23]. Those in this cohort who had attended a RLMW (n=345) and were partnered (n=280) were randomised and emailed in groups of 10"

My apologies if this is not clear. Attempts to be concise and to limit words make these somewhat complex explanations difficult. I have modified to attempt to clarify:

Participants were partners of PwMS. The PwMS were from an international online cohort of 2466 participants in the Health Outcomes and Lifestyle In a Sample of people with Multiple sclerosis (HOLISM) study [23]. Those in the HOLISM cohort who indicated in their surveys that they had attended a RLMW (n=345) and that they were partnered (n=280) were selected from the dataset. These people with MS were then randomised and emailed in groups of 10 and asked to forward the email to their partner if agreeable. The email invited the partner to provide consent to participate and complete a short survey of demographic data, details of the workshop the person with MS (and potentially they) attended and names of facilitators if known. A “No” response from the partner concluded the survey. Reasons for declining participation were not explored. A “Yes” response led to contact by researchers to organise an interview with the partner. Interviews and analysis occurred concurrently, enabling researchers to identify when apparent data saturation occurred [24] and, at that point, further recruitment ceased.

3) In conclusion, It is not clear what does it means the follow.Our study found that partners of PwMS who undertake lifestyle modification as part of MS management, experience some doubts and uncertainties common to those of other partners dealing with MS. However, lifestyle modification and its observable outcomes enabled some partners to develop a sense of empowerment and control, and a subsequent confidence and positivity about the future. Such perspectives are novel and add support for the recommendation of modification of lifestyle-related risk factors in MS. Clinicians should consider lifestyle modification in the management of PwMS and their partners."

I have made the following clarification:

In the discussion, I have added:

Some of the themes reported by our partners were similar to themes described in the literature by partners of PwMS who adopted conventional medical management. This was particularly true of the uncertainty that the unpredictable course of MS caused. Despite the PwMS having engaged with lifestyle modification, some partners still experienced uncertainty about what their futures held.

And in the Conclusion I have added:

Partners in our study experienced some doubts and uncertainties common to those of other partners dealing with MS. However, lifestyle modification and its observable outcomes for the PwMS, enabled some partners to develop a sense of empowerment and control, and a subsequent confidence and positivity about their futures. Such perspectives are novel and add support for the recommendation of modification of lifestyle-related risk factors in MS. Clinicians should consider lifestyle modification in the management of PwMS and their partners.

Round 2

Reviewer 2 Report

The authors satisfactorily responded to my previous comments. I have one (minor) further comment: I suggest moving lines 102-105 and 107-108 into section 2.6 Analysis

Reviewer 3 Report

The authors are addressed all my suggestions